# Interfacial Charge Transfer Effects of MoS_2_/α-Fe_2_O_3_ Nano-Heterojunction and Efficient Photocatalytic Hydrogen Evolution under Visible-Light Irradiation

**DOI:** 10.3390/nano13202763

**Published:** 2023-10-15

**Authors:** Tsung-Mo Tien, Edward L. Chen

**Affiliations:** Coastal Water and Environment Center, College of Hydrosphere Science, National Kaohsiung University of Science and Technology, Kaohsiung 81157, Taiwan

**Keywords:** heterojunction, photocatalytic hydrogen evolution (PHE), MoS_2_/α-Fe_2_O_3_

## Abstract

Researchers have made efforts to develop high-productivity photocatalysts for photocatalytic hydrogen production to reduce the problem of a lack of energy. Bulk semiconductor photocatalysts mainly endure particular limitations, such as low visible light application, a quick recombination rate of electron–hole pairs, and poor photocatalytic efficiency. The major challenge is to improve solar-light-driven heterostructure photocatalysts that are highly active and stable under the photocatalytic system. In this study, the proposed nano-heterojunction exhibits a great capacity for hydrogen production (871.2 μmol g^−1^ h^−1^), which is over 8.1-fold and 12.3-fold higher than that of the bare MoS_2_ and bare α-Fe_2_O_3_ samples, respectively. It is demonstrated that the MoS_2_/α-Fe_2_O_3_ heterojunction gives rise to an enhanced visible light response and accelerated photoinduced charge carrier separation. This work provides an improved visible light absorption efficiency and a narrowed energy band gap, and presents a “highway” for electron–hole pairs to promote transfer and inhibit the combination of photoinduced charge carriers for the utilization of nano-heterojunction photocatalysts in the field of hydrogen production.

## 1. Introduction

Solar-driven photocatalytic hydrogen evolution (PHE) via water splitting has garnered increasing attention because of the remarkably favorable and useful strategies used to transform renewable solar light into neat chemical energy [1,2]. Because it is essential that supportable growth diminishes the movement of carbon dioxide (CO_2_) and organic compounds to their surroundings, practical nanoparticles for novel clear energy utilization processes like photocatalysis and electrocatalysis routes have been effectively probed for decades [3]. Visible-light-driven nanomaterials are used to split water and transform solar light energy into purified energy [4,5]. Within all observed semiconductor photocatalysts, all efforts to produce hydrogen (H_2_) through heterojunction-based photocatalysis to date have unfavorably fallen short of the capability rates needed for efficient utilization [6]. To figure out this issue and obtain sustainable growth, renewable energy sources that could be employed and offered as the main renewable power source, like H_2_, need to be evaluated [7]. Nearly all scientists have taken advantage of its facilely managed catalyst configuration [8,9] and handily managed the band gap energy to regularly realize it as one of the most favorable semiconductors. Photocatalytic H_2_ evolution from water with visible light activity has attracted much attention owing to its achievable utilization in the transition and capacity of abundant solar sources. Cocatalysts commonly act as reactive locations for the production of photoinduced carriers and decrease the response energy barriers for photocatalytic H_2_ production. Among these semiconductor photocatalysts, hematite (α-Fe_2_O_3_) has great photocatalytic efficiency for hydrogen generation [10]. The photo-Fenton photocatalyst is broad because of its proper energy band gap (~2.1 eV), inexpensiveness, high stability, and environmental kindness [11,12]. The photo-Fenton reaction is a procedure in which Fe^2+^ and H_2_O_2_ join to generate ^•^OH. On the other hand, the photocatalytic efficiency of α-Fe_2_O_3_ is constrained by its short diffusion length, high electron–hole recombination rate, and the poor electrical conductivity of charge carriers [13]. The suggested MoS_2_/α-Fe_2_O_3_ (MF) nano-heterojunction exhibits a greater H_2_ generation rate of 4356 μmol h^−1^ g^−1^ (20 mg sample), which corresponds to that of pristine MoS_2_ and α-Fe_2_O_3_. In addition, the MF nano-heterojunction displays superior stability and recyclability in the 5th cycle test of photocatalytic hydrogen generation.

So far, with regard to traditional ZnO and TiO_2_ photocatalysts, particular photocatalysts like MoS_2_, Co_3_O_4_, Cu_2_O, WS_2,_ and Fe_2_O_3_ have been more broadly investigated in the area of photocatalysis [14,15]. MoS_2_, as the favorable substitute, has garnered great attention owing to its abundance and the suitable free energy of the H_2_ production rate [16]. Nowadays, MoS_2_ is known to be a good semiconductor for photocatalytic H_2_ generation with visible light activity [17]. The photocatalyst has a broad range of solar light absorption abilities, has a high capacity for photoinduced charge carrier separation, and a high transfer ability, meaning that it can meet the terms of high-performance interfacial photocatalysis [18]. The great photoinduced charge carrier migration capacity, high H_2_ generation rate and excellent photocatalytic stability of MoS_2_-based semiconductors make them favorable for photocatalytic H_2_ generation. In prior studies, phase junctions on the same material have displayed good photocatalytic efficiency because of their particular benefits, including involving elements, having good electron affinity, and vigorous coupling [19,20].The confined active locations and the unfortunate inherent conductibility of MoS_2_ prohibit the whole process of PHE. Therefore, integrating the advantages of great conductivity and abundant active sites is still a strenuous mission. Various fabrication and treatment methods have been presented for altering the band gap energy in order to collect solar light and enhance the electron–hole pair separation efficiency in photocatalysis [21,22]. Between these materials, the utilization of a co-catalyst has attracted immense interest due to the heterojunction structure, which could be useful regarding the removal rate of photoexcited electrons and when aiming to avoid the surface electron–hole pair recombination of bulk material [23]. As one of the significant photocatalysts, α-Fe_2_O_3_ has excellent properties, a good capacity for visible photoelectric response and a large specific surface area [24,25]. Meanwhile, α-Fe_2_O_3_ has great electrical and optical properties, and superior utilization in the fields of photocatalytic and environmental science. 

To overcome the above problems, several procedures have been used to improve the photocatalytic efficiency by enhancing the charge carrier separation rate; these include the addition of chemicals to alter the energy band gap of heterojunctions with cocatalysts and the building of Z-scheme routes. Semiconductor heterojunctions have been suggested to improve interfacial electron–hole pair transfer and the separation of charge carriers, driven by an internal electrostatic field in the heterojunction structure. Consequently, it is beneficial to create photocatalyst nano-heterojunctions that not only improve the practical electron and hole separation rate, but also promote their transfer efficiency. Up to now, Fe_2_O_3_/MoS_2_ heterostructure photocatalysts have been developed and studied for the degradation of various organic compounds, such as a Fe_2_O_3_/MoS_2_ heterojunction for the photo-Fenton reaction [26,27], and a Fe_2_O_3_/MoS_2_ nanocomposite and QDs for the degradation of dye [28,29]. Moreover, nanocomposites can be quickly formed to form intimate contact heterostructures with different kinds of semiconductors. When two materials come into contact, the charge carriers around the interface of the two materials diffuse through the interface, owing to the density of the charge carrier elevation. Furthermore, nano-scale photocatalysts are more easily transferred from the inner part to the surface because of their small dimensions, which make it much less probable that charge carrier recombination will occur [30,31]. In addition, Chen [32] proposed a novel Z-scheme Fe_2_O_3_/MoS_2_ nanohybrid, in which potential photoinduced electron–hole pairs are saved in the conduction band of the nanohybrid, while inhibition photoexcited electrons and holes are recombined under visible light irradiation. Therefore, the formation of nano-heterojunctions via the merging of narrow-energy-band-gap MoS_2_ nanoparticles with other nanomaterials with more positive acceleration leads to the separation and transfer of photoexcited carriers for photocatalytic hydrogen evolution.

In this case, the fabricated MF nano-heterojunction will promote the transfer of electrons and inhibit the electron–hole pair recombination rate on the surface of the nanocomposite, as well as contribute great reactive sites for photocatalytic H_2_ generation. Herein, these heterojunctions not only provide abundant active sites to improve the capacity for photocatalysis, but also inhibit the recombination efficiency of photo-response charge carriers on the surface of the photocatalyst [33]. Furthermore, an appropriate photocatalysis mechanism was presented depending on the active sites catching test, ESR analysis and photoelectrochemical experiments. This study provides guidance for the design and formation of Z-scheme heterojunction photocatalysts toward photocatalytic H_2_ production.

## 2. Materials and Methods

Sodium molybdate dihydrate (Na_2_MoO_4_·2H_2_O), thiourea (CH_4_N_2_S), iron(III) nitrate nonahydrate (Fe(NO_3_)_3_·9H_2_O), and hydrochloric acid (HCl, 36.5%) were adopted as precursors for the fabrication of MoS_2_ and α-Fe_2_O_3_, respectively, and purchased from Sigma Aldrich. All the chemical reagents used in this study were provided without further purification.

As shown in Figure 1, the MF nano-heterojunction used in this work was synthesized with a hydrothermal process [34], in which CH_4_N_2_S (2.5 mmol) and Na_2_MoO_4_·2H_2_O (1.5 mmol) were mixed and sonicated evenly in 40 mL of water and kept at 30 °C for 30 min (solution 1). Fe(NO_3_)_3_·9H_2_O was dissolved in 40 mL of distilled water and kept at 30 °C for 30 min (solution 2). The pH value of the mixture solution was set to 7. The mixture solution was distributed constantly using an ultrasonic process, stirred extensively for 30 min, then delivered into a 100 mL Teflon-lined stainless steel autoclave and kept at 200 °C for 24 h. After the hydrothermal process, the sample was cooled down naturally and washed with diluted HCl, H_2_O, EtOH and NMP three times. Then, the products were dried at 70 °C under vacuum overnight. The adding content of α-Fe_2_O_3_ was easily altered by the addition of Fe(NO_3_)_3_·9H_2_O. The nominal amounts of Fe(NO_3_)_3_·9H_2_O added to the mixture solution were 0.5, 1, 1.5, and 2 mmol, and the ratios of α-Fe_2_O_3_ for the received samples were MF-1, MF-2, MF-3, and MF-4, respectively.

The photocatalytic tests were performed in a 50 mL glass reactor equipped with a 350 W Xe lamp and a UV cut-off filter (λ > 420 nm). Firstly, the 20 mg photocatalytic product was distributed in a 50 mL mixture solution of 0.25 M/0.35 M Na_2_SO_3_/Na_2_S and lactic acid of 10% sacrifice reagent into a quartz bottle reactor. Before illumination, the suspension of the photocatalyst was distributed using an ultrasonic conductor for 20 min, and thus degassed via the bubbling of N_2_ for 30 min. The extent of photocatalysis H_2_ production was evaluated using gas chromatography (GC 7890, TCD detector).

The crystal structure of the photocatalytic materials was studied using X-ray diffraction (XRD) on a Bruker D8 Advanced with Cu Kα radiation. The morphologies of the product were displayed using field emission scanning electron microscopy (FE-SEM, ZEISS AURIGA, Tokyo, Japan). Transition electron microscopy (TEM) observations were characterized with a TEM, JEOL JEM-2100F (Tokyo, Japan) electron microscope. The elemental mapping and composition were collected using an energy-dispersive X-ray spectrometer and evaluated using TEM analysis. The UV–vis spectra (UV–vis) were determined on a spectrophotometer (Hitachi UV-4100, Tokyo, Japan) in the range of 300–800 nm to record the optical response of the prepared samples. The photoluminescence (PL) emission spectrum of the products was recorded using a spectrophotometer (F-7000 FL, Tokyo, Japan), with a xenon lamp acting as the visible light source in the PL spectrophotometer. The elemental compositions and chemical states were recorded using X-ray photoelectron spectroscopy (XPS, Thermo ESCALAB, UK). Electron spin resonance (ESR) analysis was performed using a spectrometer (JEOL JES-FA200, Tokyo, Japan) under visible light activity.

## 3. Results

XRD analysis was performed to check the crystal and phase structure of the as-fabricated samples and the received results are exhibited in Figure 1. The diffraction peaks located at 14.5°, 32.3°, 39.2°, 44.3° and 60.2° are assigned to the (002), (100), (103), (006) and (110) planes, which are properly matched with the crystal phase of MoS2 and correspond to the JCPDS card (JCPDS No. 73-1503) [35]. The XRD analysis suggests that the semi-crystalline behavior of the fabricated MoS_2_/α–Fe_2_O_3_ nano-heterojunction and events are beneficially matched with the results received. The feature peaks are comparable with the anatase phase of α-Fe_2_O_3_. Several additional feature peaks located at 2θ of 24.4°, 33.6°, 35.6°, 49.3°, 54.2°, and 64.5° correspond to the (012), (104), (110), (116), (214), and (300) planes (JCPDS No. 79-0007) [36] of the rhombohedra phase of α-Fe_2_O_3_, respectively, in the heterojunction. Owing to the covering of the peaks of MoS_2_ and α-Fe_2_O_3_, some slight altering and extending of the feature peaks were noted. Divided peaks comparable to MoS_2_ in the XRD analysis of the MoS_2_/α-Fe_2_O_3_ nano-heterojunction were not found, which could be owing to the presence of a low amount. The findings regarding MoS_2_ do not change the structural and crystallinity characteristics of the heterostructures. The gradual decrease in the peak intensity as the content of α-Fe_2_O_3_ increased apparently implies that α-Fe_2_O_3_ was successfully loaded on the binary MF nano-heterojunction. When α-Fe_2_O_3_ is combined with the MF nano-heterojunction, the diffraction feature peak of α-Fe_2_O_3_ does not emerge clearly. This is possibly due to the low amount of α-Fe_2_O_3_ and the well-featured peak of the MF nano-heterojunction that covers the diffraction feature peak of α-Fe_2_O_3_. In addition, the XRD analysis results demonstrate that the feature peak of the MF nano-heterojunction gradually diminishes with the increase in the α-Fe_2_O_3_ amount. This could be due to the addition of α-Fe_2_O_3_ having a respective influence on the crystal structure of the MF nano-heterojunction, proving that the nano-heterojunction was successfully fabricated.

To confirm the optical properties of the fabricated photocatalysts, the products were measured using a UV–vis spectrometer in the 300–800 nm wavelength range. Figure 1b exhibits the UV–vis measurement of MoS_2_, α-Fe_2_O_3_ and the MF nano-heterojunction. It was evident that the heterojunction had a stronger capacity to absorb light in the visible light range than MoS_2_ and α-Fe_2_O_3_, which could be attributed to the α-Fe_2_O_3_ experiencing more significant absorbance in the range (λ > 450 nm) than the MoS_2_ samples. The change in absorbance from the UV to visible range extends with a rise in the amount of α-Fe_2_O_3_ added. A decrease in the energy band gap of the MF nano-heterojunction was observed with the addition of α-Fe_2_O_3_ over the MF nano-heterojunction. The low energy band gap of the heterostructures illustrated the efficient excitation under visible light activity and the improved photocatalytic performance of the MF nano-heterojunction. The improved absorption of the photocatalyst means that many photons are absorbed, inducing more charge carriers in the photocatalyst and promotes the efficiency of H_2_ evolution. As can be observed in Figure 1c, the forbidden band value of the bare α-Fe_2_O_3_ was calculated as 2.31 eV; in addition, the band gap values of the MF-1, MF-2, MF-3, and MF-4 were 2.28, 2.25, 2.21 and 2.15 eV, respectively. Then, the heterostructure developed via the addition of α-Fe_2_O_3_ alters the band gap value of the MF nano-heterostructure photocatalyst. The analysis results demonstrate that the energy band gap of the MF nano-heterojunction decreased as the amount of α-Fe_2_O_3_ sample increased, suggesting that the introduction of the Fe component into the MoS_2_ lattice altered the band gap levels.

The morphologies and dimensions of the MoS_2_, α-Fe_2_O_3_, and MF nano-heterojunction are presented in SEM (Figure 2a–c) images. MoS_2_ displayed a granular stacked structure (Figure 2a), and α-Fe_2_O_3_ displayed a sphere structure (Figure 2b). The heterojunction MF photocatalyst obtained by joining MoS_2_ with α-Fe_2_O_3_ is illustrated in Figure 2c. Several clusters of particles for the heterojunction were further obtained. Less clustered nanomaterials influence the morphology and dimension of the nanostructure [37]. The TEM test of the characteristic MF-3 nano-heterojunction that displayed the greatest photocatalytic efficiency is exhibited in Figure 2d. The TEM analysis showed that the nanomaterials of the fabricated photocatalysts were consistently distributed and sphere-shaped in nature, with dimensions of 6–18 nm. Furthermore, it can be noted from the HRTEM analysis (Figure 2e) that the MoS_2_ and α-Fe_2_O_3_ material form a great heterojunction interface mode, which promotes the charge transport between photocatalysts and thus might improve the capabilities of the heterojunction photocatalysts. The interplanar lattice spacing of the heterojunction, which results in lattice fringe spacings of 0.26 nm and 0.25 nm in the MF-3 nano-heterojunction, belongs to the (002) planes of MoS_2_ and the (104) planes of α-Fe_2_O_3_, respectively. Figure 2f–i shows the EDS mapping of the MF nano-heterojunction. The EDS mapping spectrum of the MF-3 nano-heterojunction establishes the existence of Mo, S, Fe, and O elements, and corresponds to the results of the XPS spectrum. The mapping measure confirmed the construction of the MF photocatalyst with the constant distribution of the elements in the nano-heterojunction.

XPS measurements were recorded to check the elemental components and the chemical states of the elements in the heterojunction product displaying the greatest photocatalysis performance. Therefore, the XPS survey spectrum of the MF nano-heterojunction is exhibited in Figure 3a, which confirms the existence of four elements, including Mo, S, Fe, and O in the sample. The peak position was observed to investigate the chemical state and elements of the product. The fine spectra of Mo 3d are shown in Figure 3b. The Mo in the MF nano-heterojunction could be split into two feature peaks at 232.84 and 229.71 eV, which could be ascribed to Mo 3d3/2 and Mo 3d5/2, respectively [38]. Displayed in Figure 3c are the fine spectra of S 2p, split into two feature peaks at 162.56 and 163.69 eV, which are ascribed to S 2p3/2 and S 2p1/2, respectively [35]. Figure 2d shows that the two significant peaks of Fe 2p1/2 and Fe 2p3/2 were balanced and ascribed to binding energies of 724.63 eV and 711.51 eV, respectively, which verified the good recombination rate of MoS_2_ and α-Fe_2_O_3_. In addition, there was one weak stray peak that was obtained at 719.23 eV, which could have been acquired from trace Fe3+ components [31]. As displayed in Figure 3e, the distinguished electron spectra peaks of O1s at 530.38 eV, 531.25 eV and 532.22 eV, respectively, are ascribed to the adsorption of the chemical oxygen, hydroxyl group, and lattice oxygen of α-Fe_2_O_3_ [33]. In Figure 3f, the peaks of the Raman spectra at 406.9 cm^−1^ and 380.8 cm^−1^ are shown to be ascribed to the out-of-plane A1g and in-plane E12g vibrational situations of MoS_2_, which were in agreement with earlier studies [39]. Comparable to those for MoS_2_, both the A1g and E12g forms of MF-3 switched to lower positions (379.3 cm^−1^ and 405.7 cm^−1^, respectively). This switch was attributed to the interaction between the MoS_2_ and Fe_3_O_4_ samples that formed the Fe–S bonds noted in the XPS analysis [40]. This altered the strain stress in the lattice of MoS_2_, which was influenced by its heterojunction structure, and ultimately caused an alteration in the lattice vibrational frequency of the photocatalyst [41].

The PHE efficiency of the sole photocatalysts MoS_2_ and α-Fe_2_O_3_ is rather low, but the MF heterojunction displays a high PHE efficiency, which is due to the heterojunction having a better capacity for electron/hole transfer. The PHE activity of MF-3 in the 0.25/0.35 M Na_2_SO_3_/Na_2_S solution reaches 4356 μmol g^−1^ for 5 h, which is 8.1-fold greater than that of MoS_2_ and 12.3-fold greater than that of α-Fe_2_O_3_, respectively. As depicted in Figure 4b, the PHE efficiency of six photocatalysts with various ratios was collected in the 0.35/0.25 M Na_2_S/Na_2_SO_3_ solution, and the analysis results suggested that MF-3 had a higher PHE efficiency. To promote the study of the influence of the α-Fe_2_O_3_ co-catalyst, we checked the PHE performance of various amounts of the MF nano-heterojunction. As shown in Figure 4c, 5 mg, 10 mg, 20 mg, 40 mg and 50 mg of the MF-3 nano-heterojunction all maintained enhanced PHE efficiencies (384.5, 682.1, 871.2, 872.6, and 873.7 μmol g^−1^ h^−1^) owing to the α-Fe_2_O_3_, which suggests that electron transfer occurred. With the increased content of the added α-Fe_2_O_3_ sample, the better the electron/hole transfer performance, and the greater the PHE efficiency. When the amount of addition approaches a definite level, it will influence the application efficiency of MF-3, so MF-3 is the optimal ratio of heterojunction. The charge carrier on the surface of the photocatalyst studies the pH value of the solution, which qualitatively alters and responds directly to the charge mobility of the surface of the photocatalyst. The presence of various contents of OH^−^ or H^+^ has a comparable influence on the redox reaction of the H_2_ production system. Hence, the H_2_ production tests were performed using the MF-3 nano-heterojunction at a H_2_ production rate of 871.2 μmol g^−1^ h^−1^ and various pH values to explore the influence of pH value on the PHE efficiency. As can be observed in Figure 4d, the photocatalytic nano-heterojunction performed well with various pH values of 7, 8, 9, 10, 11 and 12, respectively; the PHE activity exhibited a trend of rising with the pH when it was altered from low to high. It can be seen that a higher amount of NaOH correlates with a poor capacity for PHE, which can be attributed to the lower number of protons in the alkaline solution. The nano-heterojunction achieved optimum PHE efficiency with pH values of 7. Based on the studies in the literature [42,43,44], it is obvious that the pH value of the solution for improved PHE activity may adapt with respect to the qualities of the heterostructure and other reaction parameters. It could be concluded that adding α-Fe_2_O_3_ to a heterojunction successfully enhances PHE efficiencies. The best PHE efficiency is found to be 871.2 μmol g^−1^ h^−1^ for the MF-3 nano-heterojunction; this is 12.3-fold greater than that of pristine α-Fe_2_O_3_ and 8.1-fold greater than that of MoS_2_. The cycle stability of the MF-3 nano-heterojunction was recorded and is shown in Figure 4e. The photocatalytic performance of the heterojunction decreased from 99.2% to 98.6, 97.4, 97.1 and 96.8% for the 2nd, 3rd, 4th and 5th cyclic tests, respectively. In five consecutive tests, the H_2_ production efficiency of production after recycling did not decrease. This slight decay in the performance of the heterojunction could be attributed to a decrease in certain mass during the reformation and prevention of some active species following its repeated use. The analysis results show that the MF nano-heterojunction is more stable under photodegradation systems, which is great for the possibility of utilization [45]. Varied sacrificial agents have a significant effect on the optimal analysis results regarding the efficiency of PHE with photocatalysts. Figure 4f shows the effect of other sacrificial reagents (Na_2_S/Na_2_SO_3_ of 0.35/0.25 M, methanol of 10%, TEOA of 15%, ethanol of 10%) on the PHE efficiency of the photocatalyst. The 5 h PHE efficiency test results show that the MF-3 photocatalyst exhibits a better PHE efficiency in Na_2_S/Na_2_SO_3_ of 0.35/0.25 M, compared to a rather better PHE efficiency in the 0.35/0.25 M Na_2_S/Na_2_SO_3_ solution. The PHE performance is found to be caused by a reduced residual sacrifice agent. The enhanced stability of the MF nano-heterojunction may be attributed to heterostructures adding in α-Fe_2_O_3_ with the stabilized configuration of MoS_2_ with visible light illumination [46]. The HER efficiency of heterojunction photocatalysts is also compared with the results and reports exhibited in Table 1. For comparison, the results demonstrated that the present study reported a higher photocatalytic HER efficiency.

On the other hand, electron spin resonance (ESR) was used to also check the production of ^•^O_2_^−^ and ^•^OH radicals with regard to the PHE efficiency. As shown in Figure 5a,b, no comparable signal characteristic peak was obtained under the dark condition. With visible light activity, the feature signal peak of DMPO-^•^O_2_^−^ and DMPO-^•^OH could apparently be received, which also suggests that ^•^O_2_^−^ and ^•^OH radicals are produced in the photocatalytic step. Following the above results, the major active sites for PHE activity were found to be ^•^O_2_^−^ radicals, while the dominant species was ^•^OH radicals. The photocatalytic activity over the MF-3 nano-heterojunction occurred via an oxygen-excited route. The X-ray diffraction analysis of the MF-3 nano-heterojunction before and after the H_2_ generation cycling test is exhibited in Figure 5c. The diffraction feature peak shape is not altered, suggesting that the photocatalyst has great stability. In addition, the MF-3 catalyst has a relatively stable PHE efficiency in the 0.35/0.25 M Na_2_S/Na_2_SO_3_ solution. After 5 cycle tests of the 25 h PHE efficiency experiment without an alternative sacrifice reagent, the rate of PHE efficiency was not substantially decayed, which implied that the MF-3 heterojunction photocatalyst has higher PHE efficiency stability in the 0.35/0.25 M Na_2_S/Na_2_SO_3_ solution. The magnetic test of MF-3 was performed in an applied magnetic field of −10–10 kOe. Figure 5d shows that the MoS_2_ was found to have nearly no magnetism; periodically, the magnetic saturation values of the MF-3 heterojunction and α-Fe_2_O_3_ were 2.1 emu/g and 7.5 emu/g, respectively. This shows that the saturation magnetization of the MF-3 nano-heterostructure was improved and that it was typical of soft magnetic semiconductors, which verified that the amount of paramagnetic α-Fe_2_O_3_ sample used was conducive to the magnetic recycling and separation of the MF-3 heterojunction.

In addition, the transfer and separation efficiencies of the electron–hole pairs of the as-fabricated products were studied and the photocurrent–time curve for estimating their H_2_ production capacity was constructed. Under the electrochemical analysis, the solution of the electrolyte adopted 0.25 M Na_2_SO_4_ (pH = 7). The greater the charge carrier separation rate, the greater the transient photocurrent response of the sample. In the work, a 350 W xenon lamp was used as the visible light source. The samples displayed altering grades of photocurrent intensity (Figure 6a). The MF-3 nano-heterojunction displayed the best intensity of photocurrent response, showing that the two bare semiconductors were developed among the heterojunction interfaces that conduct electron–hole pairs and construct valid charge migration routes, thus suppressing the recombination efficiency of charge carriers. In addition, the rate of electron transfer was also checked using electrochemical impedance spectrum (EIS) analysis. In addition, the smaller radius of the curvature of the arc suggests that more rapid charge transport at the interface of the photocatalysts occurred and that the photoinduced electron–hole pairs performed better regarding separation [47,48]. In Figure 6b, it can be observed that MF-3 has lower resistivity as a photocatalyst. The resistance of the heterojunction reduces after a small amount of α-Fe_2_O_3_ is added; this shows that the joining of α-Fe_2_O_3_ decreases the nano-heterojunction resistance, permitting the photoinduced charge carriers to further operatively transfer to the active species to contribute to the H2 evolution activity. The results of the photocurrent stability test of MF-3 were confirmed and are exhibited in Figure 6c. It can be observed that the photocurrent produced by MF-3 is the best of all the tested photocatalysts, which is in good agreement with that of the PHE system. Then, the recombination rate of the charge carriers and the electrons/holes transfer was further studied using photoluminescence (PL) analysis. In Figure 6d, it can be observed that, with the addition of α-Fe_2_O_3_ to heterojunctions, the PL intensity diminishes compared to pristine MoS_2_ and α-Fe_2_O_3_. Among the photocatalysts, the MF-3 nano-heterojunction displays the minimum intensity of PL, suggesting the lowest recombination in the photoinduced charge carriers; this was helpful when enhancing the performance of the photocatalyst for H_2_ production. When further adding the α-Fe_2_O_3_ co-catalyst, the weaker peak intensity of PL was observed on the MF-3 nano-heterojunction, implying that the electron and hole recombination rates were successfully inhibited.

The above-mentioned results establish that, in the photocatalytic system, the nano-heterojunction-structured Z-scheme MF was able to powerfully separate the photoinduced charge carrier, and possessed strong redox abilities in its electron–hole pairs. These phenomena provided the marvelous photocatalytic efficiency of the fabricated Z-scheme MF nano-heterojunction in this study. The photocatalytic H_2_ evolution mechanism of the MF nano-heterojunction is shown in Figure 7. Generally, due to the reduced dimensions of its material, the enlarged surface area of the heterojunction contributes more response active species, which facilitates the migration of photoinduced charge carriers to the surface reaction of the photocatalyst [49]. Simultaneously, the addition of α-Fe_2_O_3_ leads to alterations in the conduction band (CB) and valence band (VB), enhancing the capacity for oxidation and improving the PHE efficiency [50]. α-Fe_2_O_3_ serves as an electron and hole catching center to successfully restrain the charge carrier recombination rate and enhance the stability of the MF nano-heterojunction. Furthermore, the reaction mechanism involves the migration of the photoinduced carriers of MoS_2_ to α-Fe_2_O_3_, effectively owing to the construction of the heterojunction, and also their migration to the surface of the photocatalyst [51,52]. As a result, the MF nano-heterojunction approaches high PHE efficiency and stability with visible light illumination. 

## 4. Conclusions

In summary, we designed a new and efficient approach to making MF nano-heterojunctions and studied their utilization in promoting H_2_ production through photocatalysis. In addition, the α-Fe_2_O_3_ co-catalyst formed on the surface of the MF nano-heterojunction maintains a better H_2_ production performance (871.2 μmol g^−1^ h^−1^), which is nearly 12.3-fold better compared to pristine α-Fe_2_O_3_. The construction of the MF nano-heterojunction exhibited a smaller particle size and a greater specific surface area, and the construction of the α-Fe_2_O_3_ altered the energy band gap structure of the fabricated heterostructure and enhanced the capacity for migration and separation rate of the photoinduced electron–hole pairs. As a result, the final MF system displays outstanding photocatalytic efficiency during H_2_ production with visible light irradiation. In particular, this work exhibits a new approach to the formation of heterostructures on the surface of photocatalysts during the process of photocatalysis.

## Data Availability

The data presented in this study are available upon request from the corresponding author.

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
