# Peer review of "Interfacial Charge Transfer Effects of MoS2/α-Fe2O3 Nano-Heterojunction and Efficient Photocatalytic Hydrogen Evolution under Visible-Light Irradiation"

_nanomaterials, 2023, doi:10.3390/nano13202763_

Round 1

Reviewer 1 Report

The manuscript needs major revision to the language and style. A native English speaker should check the syntax. The synthesis and characterization of such hybrids has been widely demonstrated in recent articles. Representative examples are:

Y. Zhang et al., Separ Sci. Technol. 2016, 51, 1799.

M. B. Lejbini et al., Optik 2019, 177, 112.

S. Wang et al., J. Alloys Comp. 2019, 784, 1099.

G. Khabiri et al., Ceram. Int. 2020, 46, 19600.

F. Bagheri et al., Separ Sci. Technol. 2021, 56, 3022.

L. Jia et al., J. Alloys Comp. 2022, 923, 166293.

L. Chen et al., J. Alloys Comp. 2023, 947, 169577.

The introduction should be revised in order to make clear the goal of the study.

In lines 48-51, the authors state some data about hydrogen production rate, but is not clear which is the citation.

It is not clear to the reader which is the theoretical composition of the four prepared hybrids.

Also, it is suggested that Raman analysis should be included.

The manuscript needs major revision to the language and style. A native English speaker should check the syntax.

Author Response

The manuscript needs major revision to the language and style. A native English speaker should check the syntax. The synthesis and characterization of such hybrids has been widely demonstrated in recent articles. Representative examples are:

  1. Zhang et al., Separ Sci. Technol. 2016, 51, 1799. M. B. Lejbini et al., Optik 2019, 177, 112. S. Wang et al., J. Alloys Comp. 2019, 784, 1099. G. Khabiri et al., Ceram. Int. 2020, 46, 19600. F. Bagheri et al., Separ Sci. Technol. 2021, 56, 3022. L. Jia et al., J. Alloys Comp. 2022, 923, 166293. L. Chen et al., J. Alloys Comp. 2023, 947, 169577.

Reply: Thank you for the suggestion. We have revised the manuscript carefully and tried to avoid any grammar or syntax errors. In addition, we have asked several colleagues who are skilled authors of English language papers to check the English. We believe that the language is now acceptable for the review process. On the other hand, semiconductor photocatalysis has been becoming a potential technology to solve the environmental pollution and energy crisis. The semiconductor MoS2 and α-Fe2O3 were once considered as one of the most promising photocatalysts because of their photocatalytic activity, stability, and nontoxicity. Unfortunately, the light response range and quantum efficiency of bulk MoS2 and α-Fe2O3 limited its further application. Therefore, how to achieve highly efficient visible-light-driven photocatalysts became the key point in this research field. Recently, it has been reported that coupling two or more semiconductors can enhance the separation of photogenerated electron−hole pairs for pollutants (such as dye or organic compounds). Even though bulk MoS2 and α-Fe2O3 suffered some disadvantages, such as photocorrosion, and rapid recombination of photogenerated electron−hole pairs, it was widely used as the hybrid component of semiconductor heterojunctions. However, their band structures are not entirely appropriate, and the corresponding rapid electron-hole recombination kinetics result in low quantum efficiencies; thus photocatalytic H2 generation using these hybrids is not competitive with conventional H2 production methods. As we expected, MoS2/α-Fe2O3 nano-heterojunction composites exhibited considerable activity for photocatalytic hydrogen production under visible light irradiation. Our results demonstrated the great potential of the construction of efficient photocatalysts with spatial separation of reduction and oxidation sites of the surface nano-heterojunction structure for hydrogen production. In addition, some references have been added, and changes have been incorporated into the revised manuscript.

The introduction should be revised in order to make clear the goal of the study.

Reply: Thank you for the correction. It has been described in Page 2-3, Lines 81~88:  “ To overcome the upon questions, several procedures have been used to improve photocatalytic efficiency by enhancing charge carriers separation rate, like chemical adding to alter the energy band gap of heterojunctions with cocatalysts, and building Z-scheme routes. The semiconductor heterojunctions have been depicted to better interfacial electron-hole pairs transfer and separation of charge carriers driven by an internal electrostatic field made in the heterojunction structure. Consequently, it is mainly beneficial to create nano-heterojunction of photocatalysts that not only improve the practical electron and hole separation rate but also promote their transfer efficiency. ”

In lines 48-51, the authors state some data about hydrogen production rate, but is not clear which is the citation.

Reply: Thank you for the correction. We have been revised and described some related information in revised manuscript ( Page 8, Lines 360~362):  “ The HER efficiency of heterojunction photocatalysts is also compared with results and reports are exhibited in Table 1. For comparison, the results demonstrated that the present study has higher photocatalytic HER efficiency. ”

Table 1 The photocatalytic HER efficiency of different heterojunction-based photocatalysts.

Photocatalysts

Maximum rates

(μmol h- 1 g-1)

Light source

References

TiO2/MoSe2

401

300 W Xe lamp

35

CdS/FePc

73.01

300 W Xe lamp

36

phosphorus/CoP2 /SiO2

11.79

300 W Xe lamp

37

Ti3C2 MXene@TiO2/CuInS2

356.27

300 W Xe lamp

38

MoS2/α-Fe2O3

871.2

350 W Xe lamp

This work

35 Shi, F.; Xing, C.; Wang, X. Preparation of TiO2/MoSe2 heterostructure composites by a solvothermal method and their photocatalytic hydrogen production performance. Int J Hydrogen Energy 2021, 46(78), 38636.

36 Zhang, W.; Zhou, X.; Huang, J.; Zhang, S.; Xu, X. Noble metal-free core-shell CdS/iron phthalocyanine Z-scheme photocatalyst for enhancing photocatalytic hydrogen evolution, J. Mater. Sci. Technol. 2022, 115, 199.

37 Yang, C.; Zhu, Y.; Liu, Y.; Wang, H.; Yang, D. Ternary red phosphorus/CoP2 /SiO2 microsphere boosts visible-light-driven photocatalytic hydrogen evolution from pure water splitting, J. Mater. Sci. Technol. 2022,125, 59.

38 Yang, W.; Ma, G.; Fu, Y.; Peng, K.; Yang, H.; Zhan, X.; Yang, W.; Wang, L.; Hou, H. Rationally designed Ti3C2 MXene@TiO2/CuInS2 Schottky/S-scheme integrated heterojunction for enhanced photocatalytic hydrogen evolution, Chem. Eng. J. 2022, 429, 132381.

It is not clear to the reader which is the theoretical composition of the four prepared hybrids.

Reply: Thanks for the insightful comment. Further experiments were also performed to estimate the theoretical composition of the four prepared hybrids under various measurements (Fig. 1a, Fig.2 f-i, and Fig.3 ). In addition, the EDS mapping analysis can be used to determine the elemental composition of individual points or to map out the lateral distribution of elements from the imaged area. The spatial resolution of EDS mapping depends on the average atomic number of the elements on the photocatalyst, the energy of the electronic beam and the analysis area. Considering these, the analysis method is generally used to observe the distribution of the elements in the photocatalyst. Herein, it was therefore performed to examine the distribution of Mo, S, Fe, and O and atoms. Therefore, we have to support the statements and more specific on the experimental section in the revised manuscript.

Also, it is suggested that Raman analysis should be included.

Reply: Thanks for your suggestions. We have done support the statements and more specific on the Raman analysis in Page 6, Line 272-280: “ In Fig. 3f, the peaks of Raman spectra at 406.9 cm-1 and 380.8 cm-1 were ascribed to the out-of-plane A1g and in-plane E12g vibrational situation of MoS2, which were in agreement with earlier studies [32]. Comparable to those for MoS2, both the A1g and E12g forms of MF-3 switched to lower positions (379.3 cm-1 and 405.7 cm-1, respectively). The switch was ascribed to the interaction between MoS2 and Fe3O4 samples that made Fe–S bonds noted in the XPS analysis [33]. This altered the strain stress in the lattice of MoS2, which was influenced by its heterojunction structure, and ultimately caused the alteration in the lattice vibrational frequency of the photocatalyst [34].

 ”

  1. Ramakrishna Matte, H. S. S.; Gomathi, A.; Manna, A. K.; Late, D. J.; Datta, R.; Pati, S. K.; Rao, C. N. R. MoS2 and WS2 Analogues of Graphene, Angew. Chem. Int. Ed. 2010, 49, 4059.
  2. Zhang, Z.; Shi1, R.; Wang, F.; Wang, S.; Fu, G.; Zou, X.; Li, L.; Yu, L.; Tian, Y.; Luo, F. Separable magnetic MoS2@Fe3O4 nanocomposites with multi-exposed active edge facets toward enhanced adsorption and catalytic activities, J. Mater Sci 2021, 56, 5015.
  3. Yu, J.; Yin, W.; Zheng, X.; Tian, G.; Zhang, X.; Bao, T.; Dong, X.; Wang, Z.; Gu, Z.; Ma, X.; Zhao, Y. Smart MoS2/Fe3O4Nanotheranostic for Magnetically Targeted Photothermal Therapy Guided by Magnetic Resonance/Photoacoustic Imaging, Theranostics 2015, 5(9), 931.

Fig. 3 (f) Raman spectra of MoS2 and MF-3 photocatalysts.

Reviewer 2 Report

Interfacial charge transfer effects of MoS2/α-Fe2O3 nano-heterojunction and efficient photocatalytic hydrogen evolution under visible-light irradiation. Following revisions should be made before publication:

1.       The abstract should be made more concise, highlighting the novelty and major finding. More quantitative information should be presented than qualitative.

2.       The rational design of the α-Fe2O3 and MoS2 based materials can be explained in the abstract section with the reference of following articles:

 doi.org/10.1016/j.cej.2021.129312, doi.org/10.1016/j.cej.2021.132345, and doi.org/10.1016/j.compositesb.2022.110339

3.       Why did authors not mention anything about the photo Fenton effect of Fe2O3?

4.       JCPDS card should be updated to the XRD patterns and peak should be indexed with plane.

5.       In Figure formula for Fe2o3 is mistake. The sample name should be uniform throughout the manuscript. In whole manuscript the sample was discussed indicating MoS2/α-Fe2O3 whereas differ in figures.

6.       I wonder how, S precursor does not affect Fe2O3 sites? How can I believe Fe2O3 remains pure oxides in the presence of thiourea?

7.       XPS fitting for Fe is not correct.

8.       Can we study H2 evolution kinetics with kinetic model? 

Extensive editing of English language required

Author Response

Author's Reply to the Review Report (Reviewer 2)

Interfacial charge transfer effects of MoS2/α-Fe2O3 nano-heterojunction and efficient photocatalytic hydrogen evolution under visible-light irradiation. Following revisions should be made before publication:

  1. The abstract should be made more concise, highlighting the novelty and major finding. More quantitative information should be presented than qualitative.

Reply: Thank you for your observations and letting us know about our mistakes. As

per your suggestion the following sentence was included in the abstract of the revised

manuscript: “ Researchers have effort to develop high-productivity photocatalysts for photocatalytic hydrogen production to reduce the questions of energy lack. Bulk semiconductor photocatalysts mainly endure particular limitations, such as low visible‐light application, quick recombination rate of electron-hole pairs, and poor photocatalytic efficiency. The major challenge is to improve solar‐light‐driven heterostructure photocatalysts that are highly active and stable under the photocatalytic system. In particular, the better nano-heterojunction exhibits great hydrogen production capability (871.2 μmol g-1 h-1), which is over 8.1 folds and 12.3 folds higher than that of the bare MoS2 and bare α-Fe2O3 samples, respectively. It was demonstrated that the intimate interaction between MoS2 and α-Fe2O3 gave rise to enhanced visible light response and accelerated photoinduced charge carriers separation. This work provided improved visible-light absorption efficiency and a narrowed energy band gap and presented as a “highway” for electron-hole pairs to promote transfer and inhibit the combination of photoinduced charge carriers for the utilization of nano-heterojunction photocatalysts in the field of hydrogen production.”

  1. The rational design of the α-Fe2O3 and MoS2 based materials can be explained in the abstract section with the reference of following articles:

 doi.org/10.1016/j.cej.2021.129312, doi.org/10.1016/j.cej.2021.132345, and doi.org/10.1016/j.compositesb.2022.110339

Reply: Thanks for your valuable suggestion. We have been revised and some references has been added into the revised manuscript.

  1. Why did authors not mention anything about the photo Fenton effect of Fe2O3?

Reply: Thank you very much for your observation and valuable comment, as per your suggestion the following sentence was included in the introduction of the revised manuscript. It has been described in Page 2, Lines 45~51: “ hematite (α-Fe2O3) has great photocatalytic efficiency for hydrogen generation [7]. The α-Fe2O3 semiconductor photo-Fenton photocatalyst is broadly due to its proper energy band-gap (~2.1 eV), inexpensive, high stability, and environmental kindness [11, 12]. Photo-Fenton reaction is a procedure in which Fe2+ and H2O2 join to generate OH. On the other hand, the photocatalytic efficiency of α-Fe2O3 is confined by its short diffusion length, high electron-hole recombination rate and poor electrical conductivity of charge carriers [13].”

11 Geng, Y.; Chen, D.; Li, N.; Xu, Q.; Li, H.; He, J.; Lu, J. Z-Scheme 2D/2D α-Fe2O3/g-C3N4 heterojunction for photocatalytic oxidation of nitric oxide, Appl. Catal. B Environ. 2021, 280, 119409.

12 Alp, E.; İmamoǧlu, R.; Savac, U.; Turan, S.; Kazmanl, M.K.; Genc, A. Plasmon-enhanced photocatalytic and antibacterial activity of gold nanoparticles-decorated hematite nanostructures, J. Alloy. Compd. 2021, 852, 157021.

13 Arzaee, N.A.; Noh, M.F.M.; Halim, A.A.; Rahim, M.A.F.A.; Ita, N.S.H.M.; Mohamed, N.A.; Nasir, S.N.F.M.; Ismail, A.F.; Teridi, M.A.M. Cyclic voltammetry-A promising approach towards improving photoelectrochemical activity of hematite, J. Alloy. Compd. 2021, 852, 156757.

  1. JCPDS card should be updated to the XRD patterns and peak should be indexed with plane.

Reply: Thank you very much for your observation and valuable comment. Figure 1a and these results has been removed in the revised manuscript.

Fig. 1. (a) X-ray diffraction graphs.

  1. In Figure formula for Fe2O3 is mistake. The sample name should be uniform throughout the manuscript. In whole manuscript the sample was discussed indicating MoS2/α-Fe2O3 whereas differ in figures.

Reply: Thanks for your valuable suggestion. As per your suggestion above

mentioned comment was carried in the revised manuscript (yellow color)

  1. I wonder how, S precursor does not affect Fe2O3 sites? How can I believe Fe2O3 remains pure oxides in the presence of thiourea?

Reply: We extend appreciation for your perceptive insights. The following phrase serves as evidence to support your comment. In this study, the heterojunction photocatalysts were obtained as following hydrothermal process. The synthesis of heterogeneous photocatalysts usually relies on traditional wet chemical methods, including impregnation, ion exchange, and deposition-precipitation methods. While, the heterojunction catalytic materials synthesized by these methods often have some problems such as very complex structures and uneven distribution of active sites. These problems will significantly reduce the heterojunction performance of the photocatalyst. So based on our previous experience and most literatures, we selected hydrothermal route, the experimental results showed the active sites of the MoS2/Fe2O3 nano-heterojunction photocatalyst synthesized by hydrothermal method are relatively uniform and have relatively higher photocatalytic activity. In addition, there in no significant changes observed in the UV-vis spectrum of the as-fabricated samples. MF samples shows a narrow band gap structure that can be excited under visible light irradiation. According to the UV-vis spectra of the samples, the addition of Fe2O3 extended to 500-700 nm. Furthermore, studies demonstrated that with the addition of Fe2O3, the light absorption edge of semiconductors has slightly red shift and it was attributed to the formation of a composite heterojunction between MoS2 and Fe2O3 sample. However, in photocatalytic H2 evolution processes, light absorption capacity is not sole factor. The recombination rate and charge transfer ability are also important factors, which obviously limit raw MoS2 and Fe2O3 efficiency in this study. In XPS results of this work, the relevant peaks of Mo, S, Fe and O in heterojunction could be clearly observed. This phenomenon is a kind of strong evidence for the formation of heterogeneous junctions between MoS2 and Fe2O3. The formed heterogeneous junctions between MoS2 and Fe2O3 is proved beneficial for the separation of photogenerated electrons and holes based on PC and EIS results. In addition, to support the XRD, EDS mapping, and Raman analysis were conducted to clarify heterojunction composition on MoS2 and Fe2O3.

  1. XPS fitting for Fe is not correct.

Reply: Thank you so much for your observation and valuable comment. As per your

suggestion XPS curves were deconvoluted and included as Fig.3d in the revised manuscript.

Fig. 3. XPS spectra of MoS2/α-Fe2O3 nano-heterojunction, Fe 2p (d).

  1. Can we study H2 evolution kinetics with kinetic model? 

Reply: Thanks a lot for you good comment. The H2 evolution kinetics with kinetic model is a parameter for the sample. We asked for the help to perform this experiment. Unfortunately, there is no relevant lab around us that could do this. So we are very sorry for this and ask for your kind understanding. We will consider this in future research. Thank you.

Round 2

Reviewer 1 Report

The manuscript needs some checking, regarding the language and style. The authors stated that they have included some of the suggested references mentioned in the previous report, yet, there is not one addition. It is suggested that the authors include all the suggested articles, which study the chemistry and characterization of similar hybrids (in the Introduction part). Plus, they should add a couple of relative sentences about the prior efforts towards preparing similar hybrids.

The manuscript needs some checking, regarding the language and style

Author Response

Author's Reply to the Review Report (Reviewer 1)

The manuscript needs some checking, regarding the language and style. The authors stated that they have included some of the suggested references mentioned in the previous report, yet, there is not one addition. It is suggested that the authors include all the suggested articles, which study the chemistry and characterization of similar hybrids (in the Introduction part). Plus, they should add a couple of relative sentences about the prior efforts towards preparing similar hybrids.

Reply: Thank you for the correction. According to your constructive suggestion, the manuscript and the suggested references has been revised as follow below and it has done support the statements in Page 3, Lines 88~103: “Up to now, Fe2O3/MoS2 heterostructure photocatalysts have been developed and studied for the degradation of various organic compounds, such as Fe2O3/MoS2 heterojunction for photo-Fenton reaction [27, 28], and Fe2O3/MoS2 nanocomposite and QDs for degradation of dye [29, 30]. Moreover, nanocomposites could be quickly conducted to form intimate contact heterostructures with different kinds of semiconductors. When two materials arrive into contact, the charge carriers around the interface of the two materials diffuse traverse the interface owing to the density of charge carrier elevation. Furthermore, nano-scale photocatalysts are easier to transfer from the inner to the surface because of their small dimension, which causes it much fewer probably that charge carriers recombination will occur [31,32]. In addition, Chen [33] proposed a novel Z-scheme Fe2O3/MoS2 nanohybrids, in which potential photoinduced electron-hole pairs are saved in the conduction band of the nanohybrid, while inhibition photoexcited electrons and holes are recombined under visible light irradiation. Therefore, the formation of nano-heterojunctions by merging narrow energy band gap MoS2 nanoparticles with other nanomaterials with more positive accelerating the separation and transfer of photoexcited carriers for photocatalytic hydrogen evolution. ”

References:

  • Wang, S.; Tang, B.; Yang, W.; Wu, F.; Zhang, G.; Zhao, B.; He, X.; Yang, Y.; Jiang, J. The flower-like heterostructured Fe2O3/MoS2 coated by amorphous Si-Oxyhydroxides: An effective surface modification method for sulfide photocatalysts in photo-Fenton reaction, Alloys Compd. 2019, 784, 1099.
  • Jia, L.; Wang, C.; Liu, H.; Chen, R.; Wu, K. Preparation of defect-enriched (SiO32-, Cu2+) –α-Fe2O3/MoS2 Z-scheme composites with enhanced photocatalytic-Fenton performance, Alloys Compd. 2022, 923, 166293.
  • Bagheri, F.; Chaibakhsh, N. Efficient visible-light photocatalytic ozonation for dye degradation using Fe2O3/MoS2 nanocomposite, Separ Sci. Technol. 2021, 56, 3022.
  • Khabiri, G.; Aboraia, A. M.; Soliman, M.; Guda, A.A.; Butova, V.V.; Yahia, I.S.; Soldatov, A.V. novel α-Fe2O3@MoS2 QDs heterostructure for enhanced visible-light photocatalytic performance using ultrasonication approach, Int. 2020, 46, 19600.
  • Chen, J.; Chuang, Y.; Yang, W. D.; Tsai, K. C.; Chen, C. W.; Dong, C. D. All-inorganic perovskite CsPbX3 electrospun nanofibers with color-tunable photoluminescence and high performance optoelectronic applications, J. Alloys Compd. 2021, 856, 157426.
  • Chen, J.; Chen, C. W.; Dong, Direct Z-Scheme Heterostructures Based on MoSSe Quantum Dots for Visible Light-Driven Photocatalytic Tetracycline Degradation, ACS Appl. Nano Mater., 2021, 4, 2, 1038.
  • Chen, L.J.; Arshad, M.; Chuang, Y.; Nguyen, T. B.; Wu, C. H.; Chen, C.W.; Dong, C. D. A novel nano-heterojunction MoS2/α-Fe2O3 photocatalysts with high photocatalytic and photoelectrochemical performance under visible light irradiation, Alloys Compd. 2023, 947, 169577.

Reviewer 2 Report

Accept

Author Response

We greatly appreciate your valuable comments about our manuscript nanomaterials-2624148.